# Whole Exome Sequencing Reveals a Novel Homozygous Variant in the Ganglioside Biosynthetic Enzyme, *ST3GAL5* Gene in a Saudi Family Causing Salt and Pepper Syndrome

**DOI:** 10.3390/genes14020354

**Published:** 2023-01-30

**Authors:** Angham Abdulrhman Abdulkareem, Bader H. Shirah, Muhammad Imran Naseer

**Affiliations:** 1Center of Excellence in Genomic Medicine Research, King Abdulaziz University, Jeddah 21589, Saudi Arabia; 2Faculty of Science, Department of Biochemistry, King Abdulaziz University, Jeddah 21589, Saudi Arabia; 3Department of Neuroscience, King Faisal Specialist Hospital & Research Centre, Jeddah 21589, Saudi Arabia; 4Department of Medical Laboratory Technology, Faculty of Applied Medical Sciences, King Abdulaziz University, Jeddah 21589, Saudi Arabia

**Keywords:** salt and pepper developmental regression syndrome, epilepsy, developmental delay, GM3 synthase, short stature, saudi arabia

## Abstract

Salt and pepper developmental regression syndrome (SPDRS) is an autosomal recessive disorder characterized by epilepsy, profound intellectual disability, choreoathetosis, scoliosis, and dermal pigmentation along with dysmorphic facial features. GM3 synthase deficiency is due to any pathogenic mutation in the ST3 Beta-Galactoside Alpha-2,3-Sialyltransferase 5 (*ST3GAL5*) gene, which encodes the sialyltransferase enzyme that synthesizes ganglioside GM3. In this study, the Whole Exome Sequencing (WES) results presented a novel homozygous pathogenic variant, NM_003896.3:c.221T>A (p.Val74Glu), in the exon 3 of the *ST3GAL5* gene. causing SPDRS with epilepsy, short stature, speech delay, and developmental delay in all three affected members of the same Saudi family. The results of the WES sequencing were further validated using Sanger sequencing analysis. For the first time, we are reporting SPDRS in a Saudi family showing phenotypic features similar to other reported cases. This study further adds to the literature and explains the role of the *ST3GAL5* gene, which plays an important role, and any pathogenic variants that may cause the GM3 synthase deficiency that leads to the disease. This study would finally enable the creation of a database of the disease that provides a base for understanding the important and critical genomic regions that will help control intellectual disability and epilepsy in Saudi patients.

## 1. Introduction

Salt and pepper developmental regression syndrome (SPDRS) is an autosomal recessive neurocutaneous disorder, characterized by epilepsy, severe intellectual disability, scoliosis, and dysmorphic features, along with pigmentation on the skin [1]. SPDRS is normally due to a homozygous or a compound heterozygous mutation in the *ST3GAL5* gene at the 2p11.2 location [2]. The *ST3GAL5* gene encodes sialyltransferase-9, also known as GM3 synthase, on chromosome 2p11.2 [3]. The syndrome is characterized by the infantile onset of recurrent seizures that are refractory to antiepileptic medications, markedly delayed psychomotor development, generalized hypo- or hyper-pigmented skin macules, abnormal movements, and visual loss [4]. The disorder is most prevalent in the Amish population and is also known as Amish infantile epilepsy syndrome [5]. In an Amish family from Ohio, a nonsense mutation was reported in *ST3GAL5* that resulted in the early termination of the enzyme, called GM3 synthase, which is the first step in the synthesis of gangliosides from lactosylceramide [6]. Homozygous mutations were also identified in the *ST3GAL5* gene in the affected members of the families [1,7]. Furthermore, a compound heterozygous mutation was reported in two Korean sisters with SPDRS [8]. Similarly, another substitution was also reported in a compound heterozygous state in two siblings with SPDRS [8]. In one of the studies, 37 Amish patients with a homozygous mutation in the *ST3GAL5* gene were also reported [9]. 

In another study, four *ST3GAL5* variants in the 5-prime UTR regions were identified, leading to alternative splicing and alternative promoter utilization [10]

Gangliosides are located on plasma membranes in mammals, where they play role in recognition and signaling activities. GM3 synthase-mutated mice were not able to synthesize GM3 ganglioside. The mice were alive and had no major abnormalities, but were sensitive to insulin [11]. Yoshikawa et al. (2009) [12] identified a Sati -/- mouse that showed no startle reflex in response to various acoustic stimulations. In this study, they concluded that the SATI-mediated synthesis of GM3 in the cochlea is crucial for hearing.

In another ophthalmologic study, four Amish patients of different ages from two related sibships with GM3 synthase deficiency, all had normal slit-lamp examinations and preserved retinal function, but possessed pale optic nerves with atrophy; this further suggested that visual loss in this disorder results from cortical visual impairment and optic nerve defects [13].

In humans, all pathogenic mutations in the *ST3GAL5* gene have been associated with a variety of clinical disorders, such as epilepsy, cerebral palsy, and profound intellectual disability. SPDRS has been associated with GM3 synthase deficiency and was originally described in a family that had an intellectual disability and pigmentary changes. After that, an Amish family was reported to have epilepsy, developmental regression and quadriplegic cerebral palsy [1,6,9]. Later on, many other studies were published with additional clinical features, such as deafness, blindness, pigmentary changes, and neurocutaneous “salt and pepper”, along with growth failure [1,7,9,14]. 

There were no reports of SPDRS in families from Saudi Arabia. In this article, we report a novel variant in the exon 3 of the *ST3GAL5* gene (OMIM: 604402) in three siblings from a Saudi family with epilepsy, short stature, and developmental delay consistent with the diagnosis of SPDRS (OMIM: 609056).

## 2. Methods

### 2.1. Ethical Approval 

The ethical committee of the Center of excellence in Genomic Medicine Research, King Abdulaziz University, Jeddah, provided ethical approval (013-CEGMR-02-ETH) for this study. The experimental design and sample collections of this study were conducted under the international guidelines, as mentioned by the Declaration of Helsinki 2013. All of the phenotypical details were taken from the family and a detailed pedigree was drawn, following the standard guidelines. 

### 2.2. Clinical Details of the Patients

A consanguineous Saudi family with a clear diagnosis of hereditary epilepsy syndrome was recruited. The family was composed of three affected individuals and one unaffected individual, along with the parents (Figure 1a). The parents were first-degree cousins and had four children, one was healthy while three were affected by a profound intellectual disability, epilepsy, and dermal pigmentation, along with dysmorphic facial features. All members of the family were examined clinically, and (WES) was performed for the patient IV-1.

**Patient IV-1:** A 14-year-old female was first investigated at the age of three years old with global developmental delay, short stature, and failure to thrive. She was born without any complications, at 39 weeks of gestational age, 3.8 kg of weight. After the age of six months, her developmental concerns were noted as she had developmental issues linked with poor feeding and failure to thrive. She had no facial dysmorphic features. She was also diagnosed with seizures and retinal degeneration during the first two years of her life. An Electroencephalogram (EEG) showed bilateral and symmetric spike/wave complexes and polyspikes (conducted at the age of three years). She was unable to sit without support and could not speak even a few words and was unable to follow commands. She had an intellectual disability with a developmental quotient of three, along with dystonic cerebral palsy. On physical examination, her growth parameters were less than the 1st percentile for weight and height, and her head circumference was 52 cm (2nd percentile). 

**Patient IV-2**: A 12 year old female with severe atopic dermatitis, developmental delay, and failure to thrive. She was born at full-term with a weight of 3.6 kg. At the age of three years old, she was diagnosed with quadriplegic cerebral palsy and moderate axial hypotonia, frequent purposeless movements, and mild spasticity of the distal lower extremities. She could not talk but vocalized with no words. She was unable to follow any commands. Normal brain MRI showed qualitative diffusion tensor imaging. She was unable to sit at six years and had a severe movement disorder. Her developmental quotient was seven. She was able to feed by mouth and was dependent for all activities of daily living. Similarly to her elder sister, an EEG for episodes of shaking showed multifocal sharps but no seizures. Her growth parameters for weight were in the 1st percentile, for height in the 2nd percentile, and for head circumference in the 5th percentile.

**Patient IV-3:** A 9 year old male with normal birth after 39 weeks of gestation. At the age of nine months, he was noticed to have developmental delay as he was unable to sit, roll, or crawl. His vision and hearing impairment were tested approximately after twelve months of age. His brain MRI was normal (performed at six years of age). After the age of two years, he started pulling to a stand, cruising on furniture, and he started babbling. He had no speech but babbles and gestures. His skin was clear. An EEG showed multifocal bilateral sharps with seizures. He was able to self-feed. On physical examination, he was less than the 1st percentile for growth, weight, and height. Neurologically, the boy was in a much better condition than his sisters as he was alert but still nonverbal and did not follow commands. He had no eye contact with some spasticity in his legs but still managed to cruise with some help. 

### 2.3. Sample Collections

DNA was extracted from the subjects’ blood after signing the informed written consent, and the DNA was stored in the EDTA tubes (Roche Life Science) as conducted previously [15]. The Nanodrop^TM^ 2000/2000c spectrophotometers (Thermo Fisher Scientific Waltham, MA, USA) were used for the measurement of the concentration of DNA.

### 2.4. Whole Exome Sequencing (WES)

The WES state-of-the-art technique was used to identify the cause of disease in a Saudi family with three affected members. The IV-1 affected member’s DNA sample was used for WES by the latest Illumina NextSeq 550 by using (High-Output v2 kit) and 2 × 76 paired-end reads were sequenced on an Illumina NextSeq as previously explained [16,17]. Quality control was maintained on the Illumina sequencing platforms and the DNA templates were constructed with accurate quantitation. The library was prepared using the Illumina platform (San Diego, CA, USA) and Twist Human Core Exome library kit (San Francisco, CA, USA).

The WES reads were obtained in the FASTQ files format from the Illumina machines and then the files were converted to BAM files and finally to variant call format (vcf) files with a total number of variants obtained. To reduce the risk of false positivity, the reads with a Phred score below 20 were trimmed out and the rest were allowed. It was performed by the latest bioinformatics Illumina packages, called “bcl2fastq v2.20.0” (https://www.bioinformatics.babraham.ac.uk/projects/fastqc). Moreover, different analysis tools were used to identify the variants linked with the disease based on homozygosity, rare/novel (MAF+0.01%) frequency, heterozygosity, functional status using (Polyphen/SIFT predicted damage), genomic position, pathogenicity, and damaging effect of the protein that shows linkage with the disease and its phenotype. Single nucleotide polymorphisms (SNPs) or variants at this phase were identified at nucleotide resolution. The recognized SNPs were contrasted with the genomAD databases (https://gnomad.broadinstitute.org/), SnpEff (http://snpeff.sourceforge.net/SnpEff.html), and 1000 genome (https://www.internationalgenome.org/). Different available Bioinformatics tools were used. The identified list of variants was further filtered for allele frequencies <5.0% in the Genome Aggregation Database (gnomAD, http://gnomad.broadinstitute.org), and frameshift, nonsense, and splice-site variants in disease-related genes with a minor allele frequency ≤1.0% were observed in gnomAD. The obtained list of variants was classified based on the American College of Medical Genetics and American College of Pathologists (ACMG/AMP), along with multiple lines of computational evidence supporting the deleterious effect on the gene or gene product (evolutionary, conservation, splicing impact), etc. [18]. The variants were reported according to the human Genome Variation Society (HGVS) nomenclature (https://varnomen.hgvs.org/). Furthermore, other in silico studies were conducted for missense variants to predict the effect of amino acid substitutions on protein structures such as PolyPhen-2 (ranges between 0.0 (tolerated) and 1.0 (deleterious)) software (http://genetics.bwh.harvard.edu/pph2) and SIFT ranges between 0.0 (deleterious) and 1.0 (tolerated) (http://sift.bii.a-star.edu.sg). We followed the standard guidelines of the ACMG. The Mutation Tester (http://www.mutationtaster.org/) was used for the identification of the disease. The PhyloP (https://www.ncbi.nlm.nih.gov/pmc/articles/ PMC4702902/), 1000 Genomes database (http://www.internationalgenome.org/), and PhyloP. Sorting Intolerant Form Tolerant (SIFT) (http://sift.bii.a-star.edu.sg/)**,** and Combined Annotation Dependent Depletion (CADD) predicts a continuous phred-like score that ranges between 1 and 99, with higher values indicating more deleterious cases (https://cadd.gs.washington.edu/) The GERP++ (http://mendel.stanford.edu/SidowLab/ downloads/gerp/) and PhastCons scores represent the probabilities of negative selection and range between 0 and 1 (http://compgen.cshl.edu/phast/). SiPhy (https://omictools.com/siphy-tool) and Exome Aggregation Consortium (http://exac.broadinstitute.org/) VEST assigns the variants a score between 0 and 1, where 1 indicates that a confident prediction of a functional mutation was used.

### 2.5. Sanger Sequencing

After obtaining the results of the WES, the identified variants were verified by using the Sanger sequencing technique in the remaining available family members. For the PCR and sequencing, the sets of the targeted primer were designed by using the online primer 3 program, and the designed primers sequence was as forward primer ST3GAL5_3F: 5′-CTGCATAGCAGGCAGACTCA-3′ and reverse primer ST3GAL5_3R: 5′-GCATATGCTTGGCAATGTTT-3′. The sequencing data files were provided by the AB1 sequencing unit, and the obtained files were analyzed with the reference sequence using the BioEdit 7.2 software.

## 3. Results

### 3.1. Whole Exome Sequencing

The WES results identified a homozygous missense variant in the exon 3 of the *ST3GAL5* gene (OMIM: 604402), where c.221T>A p.Val74Glu was changed. All three of the affected siblings were homozygous for the *ST3GAL5* variant c.221T>A, while the parents, who are first-degree cousins, were heterozygous carriers (Figure 1b). Table 1 shows the list of the pathogenic variants of the gene in the literature, and public databases such as GHMD and Clinvar. To our knowledge, this variant has not been previously reported in the literature. Furthermore, the variant is novel in gnomAD exomes and 1000 genomes. There is a moderate physiochemical difference between valine and glutamic acid. The clinical and molecular assessments are consistent with the diagnosis of SPDRS (OMIM: 609056). Furthermore, the results of the in-silico tools, publicly and commercially available, were used to aid in the interpretation of the sequence variants identified in this study, as shown in Table 2. Based on the studies and our knowledge, the variant reported in this report is pathogenic and has not been reported in literature or in any public database.

### 3.2. Sanger Sequencing

The results of the Sanger sequencing analysis confirmed the novel homozygous pathogenic variants c.221T>A in the exon 3 of the *ST3GAL5* gene, causing SPDRS with epilepsy, short stature, delayed speech, and developmental delay. In all three of the affected members, IV-1, IV-2, IV-3, with T/T on both alleles, while the parents, III-1 and III-2, had heterozygous carrier A/T at the same position; this further confirms the disease segregation, as shown in Figure 1b. Moreover, the amino acid sequence alignment of the different species underlined the strong conservation of the variants at p.Val74Glu; this further explains the impact of the amino acid change after the variant, which may have led to the disease in the affected members, as shown in Figure 1c.

## 4. Discussion

The objective of the study was to identify the causative variants in all three of the affected members of the same Saudi family with epilepsy, short stature, speech delay, and developmental delay. Sialic acid-containing glycosphingolipids are a major group of molecules that contribute critical roles throughout the human body, including the brain. Gangliosides are glycosphingolipids that are highly abundant in the brain and play regulatory roles in the developing nervous system [21]. The enzyme ST3 Beta-galactoside Alpha-2,3-sialyltransferase 5 (ST3GAL5) converts lactosylceramide into GM3 ganglioside. GM3 synthase is the first enzyme involved in the biosynthesis of gangliosides [22]. Pathogenic variants in the *ST3GAL5* gene result in a complete lack of GM3 synthase activity, leading to the elimination of all of its downstream biosynthesis products. The *ST3GAL5* gene on chromosome 2p11.2 contains nine exons (coding regions in exons four to nine) and spans approximately 44 kb [23].

The first report of SPDRS traces back to Saul et al., in 1983, who reported three African-American siblings with the disorder. In these patients, the homozygous missense mutation in the *ST3GAL5* gene was identified through WES [1]. Later on, a similar homozygous c.862C>T transition in exon 6 of the *ST3GAL5* gene was reported in a consanguineous French family who had refractory epilepsy and delayed psychomotor development [7]. Furthermore, a compound heterozygous mutation where c.584G>C resulting in a p.Cys195Ser substitution and a c.601G>A transition resulting in a p.Gly201Arg in *ST3GAL5* gene, was reported in two Korean sisters with SPDRS [8]. Similarly, another c.601G>A p.Gly201Arg substitution was also reported in a compound heterozygous state in two siblings with SPDRS [8]. Lee et al., in 2016 [8], reported two Korean sisters born to unrelated parents with GM3 synthase deficiency. A detailed report was published with the natural history of GM3 synthase deficiency in 50 Amish patients [5]. In one of the studies, 37 Amish patients with the homozygous mutation p.R288X in the *ST3GAL5* gene were also reported [9].

Furthermore, 38 Amish patients with GM3 synthase deficiency were also reported [14]. Another report showed an Iranian patient with SPDRS in 2021 [24]. This disorder was not previously reported in Saudi Arabia.

The clinical features of SPDRS are best characterized by developmental delay (100%), visual detachment (60%), slow weight gain (96%), and increased (32%) or decreased (18%) muscle tone [5]. Additional features include agitation, restlessness, and poor sleeping in affected infants. Furthermore, gastroesophageal reflux (82%), constipation (92%), and involuntary movements (84%) have also been observed. The involuntary movements were in the form of chorea, athetosis, dystonia, hyperkinesia, dyskinesia, and tremor. The ‘salt and pepper’ characteristic pattern of skin dyspigmentation was seen in some patients. Several additional functions were affected, including hearing and vision. Patients survived to a median age of 23.5 years, and the leading cause of death was acute respiratory failure.

The epileptic seizure semiology reported in the cohort of Bowser et al. included generalized non-convulsive (37%), tonic-clonic seizures (48%), complex partial (20%), behavioral arrest (16%), tonic spasms (20%), epileptic spasms (12%), gelastic seizures (4%), atonic seizures (4%), and status epilepticus [5]. EEG abnormalities were also reported, with the main ones being multifocal spike-wave discharges (97%), slow, high-voltage, disorganized background (2–4 Hz, 50–400 V) (75%), absent posterior rhythm (63%), and absent sleep-wake architecture (50%).

The high rate of consanguinity in Saudi Arabia is due to the founder effect, several tribes and allelic heterogeneity. All factors lead the population towards the novel disease, causing variants along with novel autosomal recessive genetic disorders [25,26]. Further, these conditions of disease allele variations are known to occur as a result of long runs of homozygosity [27] or missense substitutions in a homozygous state [28], leading to the comprehensive inactivation of functional genes in the human genome [29]. Advanced WES is used as a diagnostic tool for the identification of related molecular defects in patients with supposed genetic disorders. The identified diagnostic yield of WES usually ranges between 25% to 35%, with a maximum yield of 40% in trio analysis [29,30,31]. The identification of these deleterious disease-causing variants will certainly help in genetic counselling, prenatal testing, and further support the development of therapeutic strategies against the disease in the population.

## 5. Conclusions

In this article, we reported a Saudi family with SPDRS (OMIM: 609056) due to a novel homozygous missense variant in the exon 3 of the *ST3GAL5* gene (OMIM: 604402) c.221T>A p.Val74Glu. The variant described in the present study widened the genetic spectrum of *ST3GAL5*-linked diseases, which will benefit addressing this disease in the future. This is the first report of this rare genetic variant related to the *ST3GAL5* gene in Saudi Arabia. This study and the previously published reports strongly recommend and support the use of state-of-the-art WES testing from proband or patients as a result of consanguineous marriages along with a strong family history.

## Figures and Tables

**Figure 1 genes-14-00354-f001:**
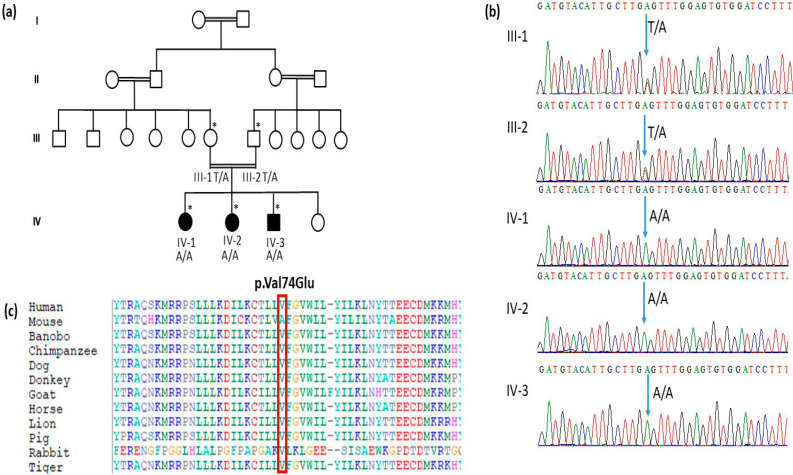
(**a**): Pedigree of the family members showing the details. (*) represent the samples available for the study. (**b**): Representative electropherogram of *ST3GAL5* gene. Sanger sequencing results showing that all three affected IV-1, IV-2, and IV-3 members had T/T on both alleles while the parents III-1 and III-2 were heterozygous carriers A/T at the same position of the *ST3GAL5* gene. (**c**): Protein alignment showed highly conserved amino acids between different species. Figure showing the conserved amino acid p.Val74Glu in *ST3GAL5* highlighted in all species.

**Table 1 genes-14-00354-t001:** Pathogenic mutations in *ST3GAL5* gene so far in literature and available database.

S. No	Mutation	Consequence	State	Origin	Condition	Reference
1	c.862C>T	p.R288 *	Homozygous	Pakistan	GM3 synthase deficiency	Gordon-Lipkin et al., 2018 [19]
2	c.584G>C c.601G>A	p.C195S p.G201R	compound heterozygous	Korean	GM3 synthase deficiency	Lee et al., 2016 [8]
3	c.1063G>A	p.E355K	Homozygous	African American	Salt and Pepper syndrome	Boccuto et al., 2014 [1]
4	c.994G>A	p.E332K	Homozygous	African American	Salt and Pepper syndrome	Boccuto et al., 2014 [1]
5	c.862C>T	p.R288X	Homozygous	French		Fragaki et al., 2013 [7]
6	c.862C>T	p.R288X	Homozygous	Old Order Amish	Amish infantile epilepsy syndrome	Simpson et al., 2004 [6] Wang et al., 2013 [14]
7	c.479C>A	p.Pro160His	Homozygous	Saudi Arabia	Salt and Pepper syndrome	Alfares et al., 2017 [20]
**8**	**c.221T>A**	**p.Val74Glu**	**Homozygous**	**Saudi Arabia**	**Salt and Pepper syndrome**	**Current study**
9	c.1255T>C	p.Ter419Arg	Homozygous	France	Intellectual disability	ClinVar
10	c.1030_1031del	p.Ile344fs	Homozygous	Iran	GM3 synthase deficiency	ClinVar
11	c.1024G>A	p.Gly342Ser	-	Saudi Arabia/USA	GM3 synthase deficiency	ClinVar
12	c.794del	p.Leu265fs	-	USA	GM3 synthase deficiency	ClinVar
14	c.369_381delinsTG	p.Lys123fs	-	USA	GM3 synthase deficiency	ClinVar
15	c.353del	p.Lys118fs	-	USA	GM3 synthase deficiency	ClinVar
16	c.332dup	p.Tyr111Ter	-	USA	GM3 synthase deficiency	ClinVar
17	c.333T>G	p.Tyr111Ter	-	USA	GM3 synthase deficiency	ClinVar
18	c.318+1del	-	-	USA	GM3 synthase deficiency	ClinVar
19	c.297T>G	p.Tyr99Ter	-	USA	GM3 synthase deficiency	ClinVar
20	c.147G>A	p.Trp49Ter	-	USA	GM3 synthase deficiency	ClinVar
21	c.124del	p.Cys42fs	-	USA	GM3 synthase deficiency	ClinVar
22	c.79C>T	p.Arg27Ter	-	USA	GM3 synthase deficiency	ClinVar
23	c.32_39del	p.Arg11fs	-	USA	GM3 synthase deficiency	ClinVar

* showing the frame shift mutation.

**Table 2 genes-14-00354-t002:** The details of in silico analysis conducted for this study. CADD-Combined Annotation Dependent Depletion; VEST (Variant Effect Scoring Tool), SIFT (Sorting Intolerant Form Tolerant).

S. No	Online Tools	Pathogenicity Score for Variant in *ST3GAL5* Gene c.221T>A p.Val74Glu
1	MutationTaster	Disease causing
2	Polyphen-2 (v2.2.2, released in Feb, 2013)	1.0
3	Mutation Assessor 2.0	1.03
4	Phylop (phyloP46way_placental)	0.72
5	VEST	0.82
6	CADD	21.0
7	Phastcons 1.4	1.0
8	SiPhy 0.5	15.0
9	Exome Aggregation Consortium Version 0.3.1	0.0%
10	1000 Genomes	0.0%
11	Diploid Internal frequency	0.0%
12	SIFT	0.09

## Data Availability

The data will be available for research on request.

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
