# Peer review of "Whole Exome Sequencing Reveals a Novel Homozygous Variant in the Ganglioside Biosynthetic Enzyme, ST3GAL5 Gene in a Saudi Family Causing Salt and Pepper Syndrome"

_genes, 2023, doi:10.3390/genes14020354_

Round 1

Reviewer 1 Report (New Reviewer)

This is an interesting work, where the authors describe a homozygous missense variant, c.221T>A, in ST3GAL5, identified for the first time in a Saudi Arabian family members, diagnosed with salt and pepper developmental regression syndrome, an autosomal recessive disorder. Although I share with the authors the importance of publishing novel variants, I have major concerns regarding the scientific approach in this case report:

1.       Extensive editing of English language and style is required

2.       Abstract:

(Lines 31-33), this sentence with the genomic information of the variant is very confusing. NM_ reference sequence should be used when referring to allelic variants for mRNA records in which a transcript corresponds to a processed sequence and, effectively, it depends of the transcription sense.

3.       Introduction

(lines 58-59): What do the authors mean that a similar homozygous mutation (in gene SIAT9) was found in a different gene (ST3GAL5)?

Introduction: the authors describe variants found in previously published cases, and this information is repeated in the discussion. I suggest that introduction should be focused on the background of gene description (e.g. gene function and structure), and the disorder. The published clinical cases are well suited in the discussion section.

4.       Methods

(lines 85-89): Blood and DNA samples collection should be along or before WES subheading.

 (line 108): not “his” because the patient is female

(line 111): what do the authors mean by “failure to thrive was added in this study”?

(line 134): “One affected member's DNA sample was used”, which proband?

All the acronyms’ abbreviations should be expanded when used for the first time (e.g HGVS)

5.       Results

What was the result for the healthy sister?

Line 184: authors used the term “patients” when referring to heterozygous carriers. Are the cousins of the index case also diagnosed with the same disorder? Or are they healthy subjects?

According to ACMG, variant c.221T>A is classified as Uncertain Significance because of its absence in GnomAD. Also, criteria PS3 refers to “Well-established in vitro or in vivo functional studies supportive of a damaging effect”, which is not the case. Of 20 pathogenic algorithms, 2 of them classify this variant as disease-causing (EVE, MutPred), 7 of them classify as Uncertain (such as SIFT) and 11 classify as benign (such as MutationTaster). I recommend the authors re-analyze the variant in order to address an accurate pathogenicity classification (I would suggest varsome https://varsome.com/, this is a helpful tool for variant analysis and interpretation).

Line 203: did the authors meant T/T or A/A since this is the mutated nucleotide?

6.       Discussion

Must be improved by also relating  the clinical cases identified in this work in the Saudi Arabian family, their clinical presentation and the clinical cases published until now.

Line 227: “was identified by…” should say the first author of the referenced work

Line 244: the word generalized is repeated

7.       References and links should be reviewed (E.g line 164, for SIFT (http://swissmodel.expasy.org).

Author Response

This is an interesting work, where the authors describe a homozygous missense variant, c.221T>A, in ST3GAL5, identified for the first time in a Saudi Arabian family members, diagnosed with salt and pepper developmental regression syndrome, an autosomal recessive disorder. Although I share with the authors the importance of publishing novel variants, I have major concerns regarding the scientific approach in this case report:

  1. 1.       Extensive editing of English language and style is required.

Response to the comments: English language of the manuscript has been improved by native English speaker to avoid grammar related mistakes.

  1. Abstract:

(Lines 31-33), this sentence with the genomic information of the variant is very confusing. NM_ reference sequence should be used when referring to allelic variants for mRNA records in which a transcript corresponds to a processed sequence and, effectively, it depends of the transcription sense.

Response to the comments: Thanks for the reviewer’s comments. We have removed the NM_reference sequence from the abstract as suggested.

  1. Introduction

(lines 58-59): What do the authors mean that a similar homozygous mutation (in gene SIAT9) was found in a different gene (ST3GAL5)?

Response to the comments: We have corrected the mistake and rewrite the sentence to make it clear. Thanks for the corrections.

Introduction: the authors describe variants found in previously published cases, and this information is repeated in the discussion. I suggest that introduction should be focused on the background of gene description (e.g. gene function and structure), and the disorder. The published clinical cases are well suited in the discussion section.

Response to the comments: As per suggestions of the reviewer’s we have changed the introduction and discussion part of the revised manuscript and added the details of the published clinical cases in the discussion section and removed it from the introduction. Thanks

  1. Methods

(lines 85-89): Blood and DNA samples collection should be along or before WES subheading.

Response to the comments: We have replaced the Blood and DNA samples collections before WES subheading as per suggestion of the reviewer’s.

 (line 108): not “his” because the patient is female

Response to the comments: Mistake has been corrected in the line 108. Thanks

(line 111): what do the authors mean by “failure to thrive was added in this study”?

Response to the comments: We have corrected the mistake. Thanks for the corrections.

(line 134): “One affected member's DNA sample was used”, which proband?

Response to the comments: The IV-1 affected member DNA sample was used for WES analysis.

All the acronyms’ abbreviations should be expanded when used for the first time (e.g HGVS)

Response to the comments: Human Genome Variation Society (HGVS) has been added in the revised manuscript. All other missed abbreviations also been added. Thanks for the corrections.

  1. Results

What was the result for the healthy sister?

Response to the comments: As the healthy sister was normal, so the parents were not interested to test her. They didn’t provide us the samples.

Line 184: authors used the term “patients” when referring to heterozygous carriers. Are the cousins of the index case also diagnosed with the same disorder? Or are they healthy subjects?

Response to the comments: We have corrected the mistake in the revised manuscript, as that is parents not patients. Thanks for the corrections.

According to ACMG, variant c.221T>A is classified as Uncertain Significance because of its absence in GnomAD. Also, criteria PS3 refers to “Well-established in vitro or in vivo functional studies supportive of a damaging effect”, which is not the case. Of 20 pathogenic algorithms, 2 of them classify this variant as disease-causing (EVE, MutPred), 7 of them classify as Uncertain (such as SIFT) and 11 classify as benign (such as MutationTaster). I recommend the authors re-analyze the variant in order to address an accurate pathogenicity classification (I would suggest varsome https://varsome.com/, this is a helpful tool for variant analysis and interpretation).

Response to the comments: We have re-analyze our data and we are convinced that the identified mutation is disease causing according to the MutationTaster.  

Furthermore, according to the Vaesome as suggested by the reviewers the identified variants come under the Uncertain Significance because of its absence in GnomAD. Variant not found in gnomAD genomes, good gnomAD genomes coverage = 30.3.

Also, criteria PS3 refers to “Well-established in vitro or in vivo functional studies supportive of a damaging effect.

Hope it will be clear to the reviewer. Thanks

Line 203: did the authors meant T/T or A/A since this is the mutated nucleotide?

Response to the comments: All three affected IV-1, IV-2, IV-3 members having T/T on both alleles while the parents III-1, III-2 were heterozygous carrier A/T at the same position. Hope it will be clear. Thanks

  1. Discussion

Must be improved by also relating  the clinical cases identified in this work in the Saudi Arabian family, their clinical presentation and the clinical cases published until now.

Response to the comments: There were no reports of SPDRS in the families from Saudi Arabia. In this article, we for the first time reporting a novel variant in exon 3 of the ST3GAL5 gene (OMIM: 604402) in three siblings from a Saudi family with epilepsy, short stature, and developmental delay consistent with the diagnosis of SPDRS (OMIM: 609056). All the available clinical information already added in the manuscript. Thanks

Line 227: “was identified by…” should say the first author of the referenced work

Response to the comments: We have corrected the mistake. Thanks

Line 244: the word generalized is repeated

Response to the comments: We have deleted the repeated word in the revised manuscript.

  1. References and links should be reviewed (E.g line 164, for SIFT (http://swissmodel.expasy.org).

Response to the comments: We have corrected the links in the revised manuscript. Thanks for the valuable comments and corrections to improve the manuscript. All the changes made in the revised manuscript as track change to make it clear.

Reviewer 2 Report (New Reviewer)

I am very grateful to the editor for giving me the opportunity to review the manuscript. The article is well written, but some issues need to be addressed.

The abstract should be structured.

The keywords are intended to increase the visibility of the article. To this end, do not repeat the same words in the title and in the keywords.

The introduction is well developed. However, the objective should be specified at the end of the introduction.

Check abbreviations throughout the manuscript. Sometimes abbreviations are used that have not been explained earlier in the text.

Standardise the format of the tables.

Table captions should contain all the abbreviations used in the table.

Improve the resolution of the image of the figure.

The discussion should start with the objective of the study.

Some sentences in the discussion should be shortened to improve comprehension.

References are up to date and correct.

Author Response

Response to the Reviewer’s comments 2

I am very grateful to the editor for giving me the opportunity to review the manuscript. The article is well written, but some issues need to be addressed.

We are thankful for the reviewer’s positive comments to improve the manuscript.

The abstract should be structured.

Response to the comments: Thanks for the suggestions we followed the author’s information provided by the journal.

The keywords are intended to increase the visibility of the article. To this end, do not repeat the same words in the title and in the keywords.

Response to the comments: Thanks for the suggestions we have changed the key words as per suggestion.

The introduction is well developed. However, the objective should be specified at the end of the introduction.

Response to the comments: The objective of the study was added at the end of the introduction as suggested.

Check abbreviations throughout the manuscript. Sometimes abbreviations are used that have not been explained earlier in the text.

Response to the comments: We have checked the abbreviation throughout the manuscript and explained in the text. Thanks for the suggestion.

Standardise the format of the tables.

Response to the comments: The formatting team is responsible for the formatting of the table by the journal. Hope they will maintain the stander. Thanks

Table captions should contain all the abbreviations used in the table.

Response to the comments: All the abbreviations were added in the table. Thanks for the corrections.

Improve the resolution of the image of the figure.

Response to the comments: The figure resolutions was improved to maximum. Thanks

The discussion should start with the objective of the study.

Response to the comments: The Objective of the study has been added in the start of the introduction as suggested by the reviewer.

Some sentences in the discussion should be shortened to improve comprehension.

Response to the comments: We have revised the manuscript to avoid any confusion. Thanks

References are up to date and correct.

Response to the comments: Thanks for the positive comments. All the changes made in the revised manuscript as track change to make it clear. Thanks for the corrections to improve the manuscript. 

Reviewer 3 Report (New Reviewer)

The authors reported a novel mutation identified by WES on a previously reported gene in a Saudi family, which may potentially help with disease diagnosis for other patients.

In Table 1, the authors summarized features of other reported mutations. It might be helpful if the authors could also include an extra column to describe patient symptoms, or to score the severity of patients. As if any mutation (e.g., R288*) results in a truncated protein, patients may have symptoms with different severities.

There are minor language mistakes, for example, on lines 29-30, should be "which encodes". The authors should double-check their manuscript.

Author Response

Reviewer’s comments 3

The authors reported a novel mutation identified by WES on a previously reported gene in a Saudi family, which may potentially help with disease diagnosis for other patients.

In Table 1, the authors summarized features of other reported mutations. It might be helpful if the authors could also include an extra column to describe patient symptoms, or to score the severity of patients. As if any mutation (e.g., R288*) results in a truncated protein, patients may have symptoms with different severities.

Response to the comments: Table 1 has been changed as suggested by the editor by adding all the pathogenic variants from literature, HGMD and ClinVar database. Please see the revised table. Already we have described the disease in the column of conditions. Thanks

There are minor language mistakes, for example, on lines 29-30, should be "which encodes". The authors should double-check their manuscript.

Response to the comments: Language mistakes has been improved. Line 29-30 correction done as suggested by the kind reviewer. Thanks for the corrections. 

Reviewer 4 Report (Previous Reviewer 1)

1)     Manuscript can be improved.

2)     The scheme model can be improved.

3)     Minor spacing and spelling mistakes etc. need correction.

4)     Endnote should be used for citation and references.

Author Response

Reviewer’s comments 4

  • Manuscript can be improved.

Response to the comments: Manuscript has been improved after reviewer’s comments and corrections from reviewer’s and editor as well. Thanks

  • The scheme model can be improved.

Response to the comments: We have improved manuscript as suggested by the kind reviewers.

  • Minor spacing and spelling mistakes etc. need correction.

Response to the comments: Minor spacing and spelling mistakes checked carefully and removed in the revised manuscript.

  • Endnote should be used for citation and references.

Response to the comments: Thanks for the suggestions. Reference already formatted according to the journal requirements. We will use in our next report.

Reviewer 5 Report (New Reviewer)

Author in this studied reported salt and pepper developmental regression syndrome (SPDRS) a disorder categorized as epilepsy, profound intellectual disability, choreoathetosis, scoliosis, dermal pigmentation along with dysmorphic facial features. GM3 synthase deficiency is due to any pathogenic mutation in ST3 Beta-Galactoside Alpha-2,3-Sialyltransferase 5 (ST3GAL5) gene, which encode sialyltransferase enzyme that synthesizes ganglioside GM3. They did WES and the results showed novel homozygous pathogenic variants c.221T>A, where p.Val74Glu in exon 3 of ST3GAL5 gene. This representation the case report is well written with validation of the clear results. I suggest some changes.  

Comments:

1-      The latest reference should need to add in this report.

2-      Detailed phenotypical and clinical information’s of the family and affected members should be required to elaborate the case study.

3-      Further validation of the results of remaing family member is suggested for clear segregation of the disease phenotype.

4-      Detailed methodology required such ad WES protocol has described very briefly.

5-      Figure quality need to improve.

6-      Please add the latest reference related to the gene mutations.

7-      Conclusion of the study need to rewrite.

8-      Some of the typo errors were noted need to remove during revision. 

Author Response

Reviewer’s comments 5

  • The latest reference should need to add in this report.

Response to the comments: We have added the latest and missing references in the revised manuscript such as,

Alfares A, Alfadhel M, Wani T, Alsahli S, Alluhaydan I, Al Mutairi F, Alothaim A, Albalwi M, Al Subaie L, Alturki S, Al-Twaijri W, Alrifai M, Al-Rumayya A, Alameer S, Faqeeh E, Alasmari A, Alsamman A, Tashkandia S, Alghamdi A, Alhashem A, Tabarki B, AlShahwan S, Hundallah K, Wali S, Al-Hebbi H, Babiker A, Mohamed S, Eyaid W, Zada AAP. A multicenter clinical exome study in unselected cohorts from a consanguineous population of Saudi Arabia demonstrated a high diagnostic yield. Mol Genet Metab. 2017 Jun;121(2):91-95.

  • Detailed phenotypical and clinical information’s of the family and affected members should be required to elaborate the case study.

Response to the comments: Detailed available phenotypical and clinical information has already been added in the manuscript. Thanks

  • Further validation of the results of remaing family member is suggested for clear segregation of the disease phenotype.

Response to the comments: All the available samples of the family members were used for Sanger Sequencing validation. Other family members were not available. Thanks

  • Detailed methodology required such ad WES protocol has described very briefly.

Response to the comments: we have added the detailed methodology in the manuscript.

  • Figure quality need to improve.

Response to the comments: Figure quality has been improved. Thanks for the suggestions.

  • Please add the latest reference related to the gene mutations.

Response to the comments: We have added the latest and missing references in the revised manuscript.

  • Conclusion of the study need to rewrite.

Response to the comments: We have concluded our study to make it clear for reader. Thanks

  • Some of the typo errors were noted need to remove during revision. 

Response to the comments: All the typo error were removed in the revised manuscript. Thanks for the corrections.

Round 2

Reviewer 1 Report (New Reviewer)

Thank you for the authors to answer and review all the comments/suggestions. Although, I still have some comments/questions:

Abstract:

Reviewer (round 1) (Lines 31-33), this sentence with the genomic information of the variant is very confusing. NM_ reference sequence should be used when referring to allelic variants for mRNA records in which a transcript corresponds to a processed sequence and, effectively, it depends of the transcription sense.

Response to the comments: Thanks for the reviewer’s comments. We have removed the NM_reference sequence from the abstract as suggested

Reviewer (round 2): In the abstract, I suggest NM_ sequence reference should be used, and not deleted. What I suggest to be deleted are the other references: NC_000002.11:g.86074909T>A and NG_012807.1 chr2:86088401A>T. Those are correct, but when describing variants in the coding region, as in this case report, NM_ sequence only should be used.

Methods

Reviewer (round 1) (line 111): what do the authors mean by “failure to thrive was added in this study”?

Response to the comments: We have corrected the mistake. Thanks for the corrections.

Reviewer (round 2): still don’t understand because this sentence refers to the clinical phenotype, and the authors include that the samples of the patient were added in this study?

Results

Reviewer (round 1): According to ACMG, variant c.221T>A is classified as Uncertain Significance because of its absence in GnomAD. Also, criteria PS3 refers to “Well-established in vitro or in vivo functional studies supportive of a damaging effect”, which is not the case. Of 20 pathogenic algorithms, 2 of them classify this variant as disease-causing (EVE, MutPred), 7 of them classify as Uncertain (such as SIFT) and 11 classify as benign (such as MutationTaster). I recommend the authors re-analyze the variant in order to address an accurate pathogenicity classification (I would suggest varsome https://varsome.com/, this is a helpful tool for variant analysis and interpretation).

Response to the comments: We have re-analyze our data and we are convinced that the identified mutation is disease causing according to the Mutation Taster shown below. 

Reviewer (round 2): Thank you for the re-evaluation but this is not completely accurate. MutationTaster algorithm uses a Bayer classifier statistical tool, not calibrated with other in silico tools. This calibration is made in Varsome and is advised for pathogenicity variant assessment (Vikas Pejaver et al. Evidence-based calibration of computational tools for missense variant pathogenicity classification and ClinGen recommendations for clinical use of PP3/BP4 criteria,  2022)

According to AMCG guidelines (2015) “Computational (In Silico) Predictive Programs – (…) The use of multiple software programs for sequence variant interpretation is also recommended. (…) It is not recommended that these predictions be used as the sole source of evidence to make a clinical assertion.”  The authors have several in silico tools results displayed in Table 2, but this should be more detailed so the reader can easily interpret for example the meaning of score 1.0 in Polyphen.

Furthermore, according to the Vaesome as suggested by the reviewers the identified variants come under the Uncertain Significance because of its absence in GnomAD.

Variant not found in gnomAD genomes, good gnomAD genomes coverage = 30.3.

Reviewer (round 2): Not only because of its absence in gnomAD, but mainly because does not fulfill the remaining criteria for (likely) pathogenic.

Also, criteria PS3 refers to “Well-established in vitro or in vivo functional studies supportive of a damaging effect.

Reviewer (round 2): This is still not clear, what are the in vitro or in vivo studies that supports the damaging effect of c.211T>A?

Discussion

Reviewer (round 1): Must be improved by also relating  the clinical cases identified in this work in the Saudi Arabian family, their clinical presentation and the clinical cases published until now.

Response to the comments: There were no reports of SPDRS in the families from Saudi Arabia. In this article, we for the first time reporting a novel variant in exon 3 of the ST3GAL5 gene (OMIM: 604402) in three siblings from a Saudi family with epilepsy, short stature, and developmental delay consistent with the diagnosis of SPDRS (OMIM: 609056). All the available clinical information already added in the manuscript. Thanks

Reviewer (round 2): With my previous comment I meant to say that the previous published clinical cases and their clinical characteristics should be compared with your own clinical case, in Saudi Arabian family. Did you find any differences/ similarities with your case report?

Author Response

Reviewer’s comments round 2

Thank you for the authors to answer and review all the comments/suggestions. Although, I still have some comments/questions:

Abstract:

Reviewer (round 1) (Lines 31-33), this sentence with the genomic information of the variant is very confusing. NM_ reference sequence should be used when referring to allelic variants for mRNA records in which a transcript corresponds to a processed sequence and, effectively, it depends of the transcription sense.

Response to the comments: Thanks for the reviewer’s comments. We have removed the NM_reference sequence from the abstract as suggested

Reviewer (round 2): In the abstract, I suggest NM_ sequence reference should be used, and not deleted. What I suggest to be deleted are the other references: NC_000002.11:g.86074909T>A and NG_012807.1 chr2:86088401A>T. Those are correct, but when describing variants in the coding region, as in this case report, NM_ sequence only should be used.

Response to the comments: Thanks for the reviewer’s comments. We have used the NM reference sequence and removed the other references from the abstract as suggested.

Methods

Reviewer (round 1) (line 111): what do the authors mean by “failure to thrive was added in this study”?

Response to the comments: We have corrected the mistake. Thanks for the corrections.

Reviewer (round 2): still don’t understand because this sentence refers to the clinical phenotype, and the authors include that the samples of the patient were added in this study?

Response to the comments: We have corrected the mistake. Thanks for the corrections.

Results

Reviewer (round 1): According to ACMG, variant c.221T>A is classified as Uncertain Significance because of its absence in GnomAD. Also, criteria PS3 refers to “Well-established in vitro or in vivo functional studies supportive of a damaging effect”, which is not the case. Of 20 pathogenic algorithms, 2 of them classify this variant as disease-causing (EVE, MutPred), 7 of them classify as Uncertain (such as SIFT) and 11 classify as benign (such as MutationTaster). I recommend the authors re-analyze the variant in order to address an accurate pathogenicity classification (I would suggest varsome https://varsome.com/, this is a helpful tool for variant analysis and interpretation).

Response to the comments: We have re-analyze our data and we are convinced that the identified mutation is disease causing according to the Mutation Taster shown below. 

Furthermore, according to the Vaesome as suggested by the reviewers the identified variants come under the Uncertain Significance because of its absence in GnomAD.

Variant not found in gnomAD genomes, good gnomAD genomes coverage = 30.3.

Also, criteria PS3 refers to “Well-established in vitro or in vivo functional studies supportive of a damaging effect. Hope it will be clear to the reviewer. Thanks

Reviewer (round 2): Thank you for the re-evaluation but this is not completely accurate. MutationTaster algorithm uses a Bayer classifier statistical tool, not calibrated with other in silico tools. This calibration is made in Varsome and is advised for pathogenicity variant assessment (Vikas Pejaver et al. Evidence-based calibration of computational tools for missense variant pathogenicity classification and ClinGen recommendations for clinical use of PP3/BP4 criteria,  2022)

According to AMCG guidelines (2015) “Computational (In Silico) Predictive Programs – (…) The use of multiple software programs for sequence variant interpretation is also recommended. (…) It is not recommended that these predictions be used as the sole source of evidence to make a clinical assertion.”  The authors have several in silico tools results displayed in Table 2, but this should be more detailed so the reader can easily interpret for example the meaning of score 1.0 in Polyphen.

Response to the comments: We totally agree with the reviewer’s comments that MutationTaster algorithm uses a Bayer classifier statistical tool, not calibrated with other in silico tools. Furthermore, we are also thankful for the reviewer’s suggestions along with the references of the ACMG. We have added the details in the method part so the reader can easily interpret as suggested. Thanks for the comments to improve the manuscript.  

Reviewer (round 2): Not only because of its absence in gnomAD, but mainly because does not fulfill the remaining criteria for (likely) pathogenic.

Response to the comments: Yes, we agree with the reviewer’s comment that the identified variants not only absent in gnomeAD but mainly does not fulfill the criteria of likely pathogenic that is why it lies under VUS.  Identifying variants that are significant or likely to be significant is a difficult task that may require expert human and in silico analysis, laboratory experiments and even information theory to validate it. Further animal model studies and In Silico validation is always required.

Reviewer (round 2): This is still not clear, what are the in vitro or in vivo studies that supports the damaging effect of c.211T>A?

 Response to the comments: According to standard guidelines of the ACMG classification of variants that come under the PS3: Well-established in vitro or in vivo functional studies supportive of a damaging effect on the gene or gene product in previous studies such as  Yoshikawa et al. (2009); Fragaki et al. (2013). While the damaging effect of our identified variants c.211T>A is validated by using some of the In Silico tools mentioned in the Table 1.

Discussion

Reviewer (round 1): Must be improved by also relating  the clinical cases identified in this work in the Saudi Arabian family, their clinical presentation and the clinical cases published until now.

Response to the comments: There were no reports of SPDRS in the families from Saudi Arabia. In this article, we for the first time reporting a novel variant in exon 3 of the ST3GAL5 gene (OMIM: 604402) in three siblings from a Saudi family with epilepsy, short stature, and developmental delay consistent with the diagnosis of SPDRS (OMIM: 609056). All the available clinical information already added in the manuscript. Thanks

Reviewer (round 2): With my previous comment I meant to say that the previous published clinical cases and their clinical characteristics should be compared with your own clinical case, in Saudi Arabian family. Did you find any differences/ similarities with your case report?

Response to the comments: The cases had a similar phenotype to other published cases in the literature.

All the response of round 2 are highlighted red color.  

Reviewer 2 Report (New Reviewer)

The author addressed all my comments

Author Response

Thanks for the reviewer's positive comments to improve the study. 

This manuscript is a resubmission of an earlier submission. The following is a list of the peer review reports and author responses from that submission.

Round 1

Reviewer 1 Report

The Idea and data presented by author seems seem interesting. The manuscript needs revision particularly the following important points need to address.

1)      The title of the manuscript needs correction (grammar).

2)      Manuscript language need to improve., As there are many grammar etc.

      related mistakes.

3)      Manuscript should be in uniform font size and format.

4)      End Note recommended for citation and references.

Author Response

  • The title of the manuscript needs correction (grammar).

Response to the comments: Title of the manuscript has been corrected. Thanks for the suggestion.

2)      Manuscript language need to improve, As there are many grammar etc.

      related mistakes.

Response to the comments: Language of the manuscript has been improved by native English speaker to avoid grammar related mistakes.

3)      Manuscript should be in uniform font size and format.

      Response to the comments: Format and font size has been corrected. Thanks for the suggestion.

4)      End Note recommended for citation and references.

         Response to the comments: All the references has been corrected. Thanks for the suggestions.

Reviewer 2 Report

The current work if of interest for scientists working on SPDRS as it reports a novel mutation that co-segregated with the phenotype. 

The reviewer advises to add the following : 

IRB approval number.

homogenize the mutation nomenclature; either use p.R288* or p.R288X.. Need to add the transcript refseq number too (NM) in the table.

remove the word gene when the gene symbol is written in italic..

Author Response

The current work if of interest for scientists working on SPDRS as it reports a novel mutation that co-segregated with the phenotype. 

The reviewer advises to add the following : 

IRB approval number.

Response to the comments: We have the ethical approval (013-CEGMR-02-ETH) from the center of excellence in genomic medicine research King Abdulaziz University for this study.

homogenize the mutation nomenclature; either use p.R288* or p.R288X.. Need to add the transcript refseq number too (NM) in the table.

Response to the comments: Nomenclature has been corrected as suggested. NM has been added in the abstract of the manuscript.

remove the word gene when the gene symbol is written in italic..

Response to the comments: We have removed the word gene where the gene symbol written in italic. Thanks for the correction.